# GFairHint: Improving Individual Fairness for Graph Neural Networks via Fairness Hint

**Paiheng Xu** [*1], **Yuhang Zhou** [*2], **Bang An** [1], **Wei Ai** [2], **Furong Huang** [1]
[1] Department of Computer Science, University of Maryland, College Park
[2] College of Information Studies, University of Maryland, College Park
{paiheng, tonyzhou, bangan, aiwei, furongh}@umd.edu

## Abstract

Graph Neural Networks (GNNs) have proven their versatility over diverse scenarios. With increasing considerations of societal fairness, many studies focus on algorithmic fairness in GNNs. Most of them aim to improve fairness at the group level, while only a few works focus on individual fairness, which attempts to give similar predictions to similar individuals for a specific task. We expect that such an individual fairness promotion framework should be compatible with both discrete and continuous task-specific similarity measures for individual fairness, and balance between utility (e.g., classification accuracy) and fairness. Fairness promotion frameworks are generally desired to be computationally efficient and compatible with various GNN model designs. With previous work failing to achieve all these goals, we propose a novel method, **GFairHint**, for promoting individual fairness in GNNs, which learns fairness representations through an auxiliary link prediction task. The fairness representations are then concatenated with the learned node embeddings in original GNNs as *" fairness hint"*. We empirically evaluate our methods on five real-world graph datasets that cover both discrete and continuous settings for individual fairness similarity measures, with three popular backbone GNN models. The proposed method achieves the best fairness results in almost all combinations of datasets with various backbone models, while generating comparable utility results, with much less computational cost compared to the previous state-of-the-art (SoTA) model.

## 1 Introduction

Graph Neural Networks (GNNs) have shown great potential in modeling graph-structured data for various tasks such as node classification, graph classification, and link prediction [42]. Specifically, there are many real-world applications for the node classification task, e.g., recruitment [22, 44], recommendation system [24, 41, 40], and loan default prediction [13, 38]. As GNNs play important roles in these decision-making processes, there is a recent surge of increased attention to fairness in graph-structured data and GNNs [26, 8, 22]. Due to the message-passing mechanism in GNNs, where nodes learn representations by aggregating information from their neighbors, the effect of unfairness can be amplified, resulting in unexpected discrimination [9, 39, 19]. Taking social networks as an example, users tend to connect with others in the same demographic group. The message-passing mechanism may cause GNNs to perform differently for different demographic groups or similar individuals with respect to a specific task.

There are mainly two types of algorithmic fairness [27]. Group fairness attempts to *treat different groups equally*. Individual fairness, which is the focus of our work, intends to *give similar predictions to similar individuals* for a specific task. A core question for individual fairness is how to define

---
[*]Equal contribution

2022 Trustworthy and Socially Responsible Machine Learning (TSRML 2022) co-located with NeurIPS 2022.

the task-specific similarity metric. Dwork et al. [10] originally envisioned that the metric would be provided by human experts "as a (near ground-truth) approximation agreed upon by the society". Lahoti et al. [21] argues that it is very difficult for experts to measure individuals based on a quantitative similarity metric when in operationalization. They further suggest it is much easier to make pairwise judgments which results in a discrete (e.g., binary) similarity measure between two individuals. For cases where there is no task-specific similarity metric at hand, other works [29, 7, 16] use simplified notions by developing continuous similarity metrics (e.g., a weighted Euclidean distance) over a feature space of data attributes.

We highlight several desiderata for promoting individual fairness in GNNs or fairness in machine learning systems in general. (1) The proposed method for individual fairness should be compatible with both discrete and continuous similarity measures as described above. (2) Generally, we expect the models to achieve a good balance between utility (e.g., classification accuracy) and fairness when making predictions. (3) We wish that the additional computational cost introduced to promote fairness is reasonably small. (4) We hope the fairness promotion method to be compatible with different GNN model architectures and various designs for specific tasks.

In this study, we propose an individual fairness representation learning framework to improve individual FAIRness for GNNs via fairness HINT (**GFairHint**), with the above desired properties. We consider the setting in which similarity measures are available for each pair of individuals, either discrete or continuous. As shown in Figure 1, in addition to the original input graph, we create a fairness graph where the edge between two nodes is weighted by the given similarity measure and does not exist when the similarity value is $0$ or below a certain threshold. We then learn a fairness representation for each node from the constructed fairness graph via link prediction, where we encourage the model to recover randomly masked edges. The learned fairness representation is then used as a fairness hint by concatenating with the node embeddings from the original graph, which is trained in parallel to maximize utility with the main GNN model We feed the concatenated representations to multilayer perceptrons (MLPs) to make predictions.

To show the effectiveness of the proposed method, we conduct extensive empirical evaluations on five node classification datasets, with either continuous similarity measure derived from input space or discrete one provided by external annotators. We also experiment with three popular GNN backbone models, which results in 15 model $\times$ dataset comparisons in total. The proposed method achieves the best fairness results in most comparisons (12/15), with the best utility results in the 9/15 comparisons, and comparable utility performance in the other comparisons. We summarize our main contributions as follows:

- We propose a novel plug-and-play framework for promoting individual fairness in GNNs which learn fairness hint through an auxiliary link prediction task.

- The proposed method meets the above-listed desiderata for promoting fairness, as it is compatible with two different settings for similarity measures of individuals, achieves comparable accuracy while making more individually fair predictions, computationally efficient, and easy-to-integrate with different model designs.

## 2 Related Work

**Fairness for Graph-structured Data** Most previous efforts focus on promoting group fairness in graphs [32, 23, 39, 1, 4, 2], which encourages the same results across different sensitive groups (e.g., demographics). Another line of research work is on counterfactual fairness [20, 25], which aims to generate the same prediction results for each individual and its counterfactuals.

Few research studies individual fairness in graphs, more specifically fairness through awareness [10, 27]. Individual fairness intends to render similar predictions to similar individuals for a specific task. Kang et al. [16] propose a framework called InFoRM (Individual Fairness on Graph Mining) to debias a graph mining pipeline from the input graph (preprocessing), the mining model (processing), and the mining results (postprocessing), but not specifically for GNN models. Song et al. [35] identify a new challenge to enforce individual fairness informed by group equality. The work closest to ours is REDRESS [7]. They propose to model individual fairness from a ranking-based perspective and design a ranking-based loss accordingly. However, their method does not generalize well when the similarity measure is discrete, especially binary, because calculating the ranking based loss requires

to rank the individuals based on the similarity values and with discrete similarity measures (i.e., many individuals on the same similarity level) the rankings are not as informative as in the cases with continuous similarity measures. Moreover, despite their effort on reducing the computation cost and the effectiveness of the ranking-based loss, high computational cost is unavoidable when computing the rank.

**Individual Fairness** There are other works focusing on individual fairness, but not specifically for graph-structured data. The definition of individual fairness, similar predictions for similar individuals, can be formulated by the Lipschitz constraint, which inspires works such as Pairwise Fair Representation (PFR) [21] to learn fair representation as input (preprocess). Because it is computationally difficult to enforce Lipschitz constraint, Yurochkin and Sun [43] propose an in-process method with a lifted constraint and Petersen et al. [31] propose a post-processing method with Laplacian smoothing. We note that both PFR and InFoRM can be adapted to promote individual fairness with GNN models, but it is shown that REDRESS [7] largely outperforms these two methods. Thus, we only compare with REDRESS in our experiments.

## 3 Proposed Method - GFairHint

The generic definition of individual fairness is *individuals who are similar should have similar outcomes* [10]. For graph data and GNN models, we represent the similarity measure as an **oracle similarity matrix** $\mathcal{S}_F$, where the value of $(i, j)$-th entry is the similarity between the node $i$ and $j$. We assume the oracle similarity matrix is given before training, either from external annotation or input feature.

Our proposed **GFairHint** framework, as shown in Figure 1, consists of three steps. First (Section 3.1), we construct an unweighted fairness graph, $\mathcal{G}_F$, with the same set of nodes in the original input graph. The undirected edges of $\mathcal{G}_F$ represent that two nodes have a high similarity value in $\mathcal{S}_F$. Next (Section 3.2), we obtain the individual fairness hint through a representation learning method that learns fairness representations for the nodes in $\mathcal{G}_F$. Specifically, the representation learning model predicts whether two nodes in $\mathcal{G}_F$ have an edge through a GNN link prediction model whose final hidden layer output is used as the fairness hint. Finally (Section 3.3), we concatenate the node fairness hint with the learned node embedding of the GNN for original tasks and use the joint embedding for further training.

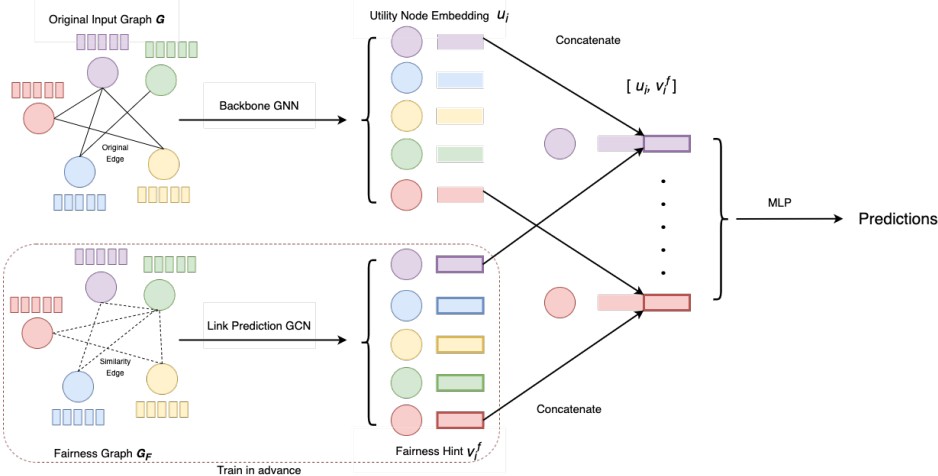

Figure 1: The proposed individual fairness promotion framework, **GFairHint**. GFairHint learns fairness hint from the fairness graph and concatenate the fairness hint with the utility node embedding from the original backbone GNN model. Finally, it feeds the concatenated node embedding into an MLP to make fair predictions. The loss function for GFairHint can be a single utility loss (cross entropy loss) or the combinations of utility loss and fairness loss (e.g., ranking-based loss).

## 3.1 Construction of Fairness Graph

The oracle similarity matrix can be constructed by incorporating various types of data sources. We show two commonly utilized data sources, i.e., external annotation and input feature.

**Oracle Similarity Matrix based on External Annotation** Constructing $\mathcal{S}_F$ is straightforward when external pairwise judgements are available on whether two individuals $i, j$ should be treated similarly given a specific task [21]. The entry $S_{ij}^F$ in $\mathcal{S}_F$ is 1 when individual $i$ and $j$ are labeled as similar, and 0 otherwise. In this case, $\mathcal{S}_F$ is the adjacency matrix for the fairness graph $\mathcal{G}_F$. An alternative type of judgements is to map individuals into discrete equivalence classes. A pair of individuals $i, j$ is linked in the fairness graph, $\mathcal{G}_F$, only if they belong to the same class.

**Oracle Similarity Matrix based on Input Feature** Although the similarity for individual fairness was originally envisioned to be provided by human experts [10], it is often impractical to obtain for real-world tasks. Previous works [7, 29] obtain the oracle similarity matrix $\mathcal{S}_F$ from input feature space i.e., the entry $S_{ij}^F$ in $\mathcal{S}_F$ is the cosine similarity between the features of node $i$ and $j$. To construct the fairness graph, $\mathcal{G}_F$, we can discretize the continuous cosine values by connecting each node only to its top-$k$ similar nodes from the oracle similarity matrix $\mathcal{S}_F$.

## 3.2 Fairness Representation Learning Model

To incorporate the fairness information into the GNN model, we learn fairness node representation with a separate GNN model using an auxiliary link prediction task on the fairness graph, $\mathcal{G}_F$. Similarly to masked language modeling in NLP to learn the contextual word embeddings by randomly masking words [6], we expect that predicting whether two nodes share a "similarity" edge inside the fairness graph can yield a valid fairness hint.

We use Graph Convolutional Network (GCN) model [18] with GCN layers for link prediction, i.e., predicting whether two nodes in $\mathcal{G}_F$ share an edge. The initial input node features of the link prediction model are the same as the features of the original task. For any node embedding $h_i^l$ (the embedding of node $i$ from the $l$th hidden layer), GCN layer combines the node embedding $h_i^l$ and other node embeddings from its neighbor node set $\mathcal{N}(i)$, which is formally denoted as

$$h_i^{l+1} = GCN(h_i^l, \{h_j^l, j \in \mathcal{N}(i)\}) \tag{1}$$

If we set the output of the last layer in the link prediction model for node $i$ and node $j$ as $v_i^f$ and $v_j^f$ respectively, the inner product $v_i^f \cdot v_j^f$ is the probability score of node $i$ and node $j$ sharing an edge. We use cross entropy loss to optimize the link prediction model.

We train this fairness representation learning model separately from the original task GNN model to avoid overfitting. We extract the output $v_i^f$ of the last layer for each node as the fairness hint.

## 3.3 Fairness Promotion for GNN Models

Our **GFairHint** framework is compatible with various GNN model architectures for the original tasks. The basic operations of each GNN layer are similar to the GCN operation in Equation 1, but the convolutional operations are replaced with other message-passing mechanisms for different GNN models. We train the chosen GNN backbone models with utility loss $\mathcal{L}_{utility}$ (i.e., cross entropy loss) and obtain the **utility node embedding** $u_i$ from the last GNN hidden layer. We then concatenate $u_i$ with the fairness hint $v_i^f$ to form a joint node embedding $[u_i, v_i^f]$. We add two layers of multilayer perceptron (MLP) with weights $W_1$ and $W_2$ to encourage the model to learn both utility and fairness information from the joint node embeddings. The final embedding $z_i$ of the node $i$ can be calculated as

$$z_i = W_2(W_1[u_i, v_i^f] + b_1) + b_2$$

We extract the fairness hint $v_i^f$ using the learned fairness representation learning model before training the original GNN model, and the fairness hint is fixed during the optimization process. For node classification tasks, we apply *softmax* to the final node embedding $z_i \in \mathbb{R}^c$ to obtain the predictions where $c$ is the number of classes and apply $\mathcal{L}_{utility}$ to optimize the parameters.

### 3.4 Integration with Fairness Loss

We can simply use the utility loss $\mathcal{L}_{utility}$ as the final loss. Moreover, we can further encourage the model to learn fairness information by adding the "fairness" loss into the final loss function. Previous fairness promotion methods have designed various fairness loss to enforce good balance between utility and fairness [7, 31], which are complementary to our fairness hint. In our work, we integrate the ranking-based fairness loss in REDRESS [7] to our framework GFairHint.

Following the procedure in REDRESS, we additionally compute an **outcome similarity matrix** $\mathcal{S}_{\hat{Y}}$ with the predicted outcome $\hat{Y}$, where the $(i, j)$-th entry is the cosine similarity between the embedding $z_i$ and $z_j$ of the node $i$ and $j$ of the final MLP layer. The general objective of the loss is to minimize the difference between the oracle similarity matrix $\mathcal{S}_F$ and $\mathcal{S}_{\hat{Y}}$. For each node, we can obtain two top-k ranking lists derived from $\mathcal{S}_F$ and $\mathcal{S}_{\hat{Y}}$ respectively. The fairness loss $\mathcal{L}_{fairness}$ is then calculated with these two ranking lists as input. The final objective is to combine the fairness loss and the utility loss.

$$\mathcal{L}_{total} = \mathcal{L}_{utility} + \gamma \mathcal{L}_{fairness} \tag{2}$$

where $\gamma$ is an adjustable hyperparameter. By changing the value of $\gamma$, we can control the weight of fairness and utility during training according to the task requirement.

## 4 Experiment Setup

**Dataset Collection**   In our work, we focus on the node classification task to evaluate the fairness promotion ability of our proposed GFairHint. We collect five real-word datasets to assess the model performance in multiple domains (see statistics in Table 3). Coauthor-CS (CS) and Coauthor-Phy (Phy) are two co-authorship network datasets [34], where each node represents an author, and they connect the nodes if two authors have published a paper together. ACM is a dataset of citation network [36], where each node represents a paper, and the edge denotes the citation relationship. These three datasets (ACM, CS, Phy) are also applied in the REDRESS paper as the experiment benchmarks [7]. In addition, we use another citation network OGBN-ArXiv dataset [14], which is several magnitude orders larger than the ACM, CS, and Phy datasets. Since the citation and co-authorship network datasets do not contained human annotated similarity, we follow previous work [7] and use the cosine similarities between node features as the entries in $\mathcal{S}_F$.

Additionally, we curate a dataset with external human annotation on individual fairness similarity in discrete setting to demonstrate our framework's compatibility when external annotation is available. The Crime dataset [33] consists of socioeconomic, demographic and law / police data records for neighborhoods in the US. We follow Lahoti et al. [21] for most of the preprocessing and introduce additional information on the geometric adjacency of the county[2] to form a graph-structured dataset. The nodes are the neighborhoods, and the edges indicate that two neighborhoods reside in the same county or adjacent counties. We have a binary outcome variable for whether the neighborhood is violent and consider other data records as input features. The details for the dataset curation and statistics are in Appendix B.

**Models for Comparison**   To show the superiority of our proposed framework, we implement the vanilla GNN models and previous SOTA as baseline models with sensitivity analysis. The baseline models are **Vanilla**, **REDRESS**, and **REDRESS + MLP**. Our models are **GFairHint** and **GFairHint + REDRESS**. Details of the model introduction are shown in Appendix D. We note that for the Crime dataset, since the entries of oracle similarity matrix $\mathcal{S}_F$ are discrete (0-1), we cannot calculate the ranking-based loss of the constructed fairness graph $\mathcal{G}_F$. Therefore, we adapt REDRESS-related models to calculate the ranking-based loss based on input feature similarity. As a result, for the Crime dataset, REDRESS and REDRESS + MLP do not have any access to the fairness information (i.e., fairness graph $\mathcal{G}_F$), while GFairHint + REDRESS get fairness information only through the fairness hint but not the ranking-based loss.

**Evaluation Metric**   Since our work focuses on the node classification task, we use conventional classification accuracy (ACC) as the metric to evaluate the utility performance of the model. Regarding the metric of individual fairness, we use different evaluation metrics in accordance with two different

---

[2]https://pypi.org/project/county-adjacency/

| Model | ACC | ERR@10 | NDCG@10 |
|---|---|---|---|
| Vanilla | 70.19 ± 0.02 | 91.45 ± 0.01 | 75.01 ± 0.19 |
| REDRESS | 68.65 ± 1.13 | 91.57 ± 0.10 | 75.25 ± 0.71 |
| REDRESS + MLP | 69.85 ± 0.27 | 91.47 ± 0.01 | 75.01 ± 0.26 |
| GFairHint | **70.62 ± 0.91** | 94.28 ± 0.09 | 81.93 ± 0.27 |
| GFairHint + REDRESS | 69.80 ± 0.47 | **95.22 ± 0.80** | **85.48 ± 3.47** |

Table 1: Node classification results on the ArXiv dataset for GCN model. All values are reported in percentage. $\mathcal{S}_F$ is defined based on the input feature, so we use ERR@10 and NDCG@10 to evaluate the fairness. The best results are in **bold**, the second best results are underlined.

| Model | ACC | Consistency |
|---|---|---|
| Vanilla | 73.83 ± 0.34 | 54.80 ± 0.23 |
| REDRESS | 73.98 ± 0.70 | 54.07 ± 0.96 |
| REDRESS + MLP | 73.58 ± 1.80 | 53.06 ± 1.04 |
| GFairHint | 75.44 ± 0.71 | 62.76 ± 2.74 |
| GFairHint + REDRESS | **75.54 ± 0.90** | **63.61 ± 4.44** |

Table 2: Node classification results on the Crime datasets for GCN model. All values are reported in percentage. $\mathcal{S}_F$ is defined based on the external annotation, so we use Consistency to evaluate the fairness. The best results are in **bold**, the second best results are underlined.

settings of the oracle similarity, i.e., continuous and discrete. For the co-authorship and citation networks (ACM, ArXiv, CS, Phy), we follow previous work [7] to utilize ERR@K [3] and NDCG@K [15] and choose $k = 10$. For the Crime dataset, where the entry of the oracle similarity matrix is discrete, we use **Consistency** [21] as the evaluation metric. The detailed description of the metrics is presented in Appendix C.

## 5  Experiment Results

**Effectiveness of GFairHint**   In this section, we present the results of our proposed methods and baselines on collected datasets. The implementation details of the experiment are shown in Appendix A. For each dataset, we choose the backbone models and model hyperparameter settings with better average utility between two large and small model size settings as described in Appendix A.

**Oracle Similarity Matrix based on Input Feature** For the citation and co-authorship networks, we use the input feature similarity as the entry of the oracle similarity matrix to construct the fairness graph. We present the results for Arxiv dataset with GCN model in Table 1. The full results are shown in Table 4 for the citation networks and Table 5 for the co-authorship networks in the Appendix. For utility performance, our proposed GFairHint and GFairHint + REDRESS models achieve comparable results with vanilla backbone GNN models and other REDRESS variation models. Indeed, in 5 cases out of 6 experiments for co-authorship datasets, our models achieve the best utility performance.

Regarding the fairness perspective, we use ERR@10 and NDCG@10 to evaluate the individual fairness promotion ability of the models, and higher values of ERR and NDCG represent better individual fairness promotion. Incorporating the fairness hint to REDRESS increases the ERR and NDCG values from 91.47 to 95.22 and from 75.01 to 85.48 respectively for Arxiv dataset with GCN model. From the results in Tables 4 and 5 (Appendix), our proposed models achieve the best fairness performance in nearly all settings, except for the ERR value of the GNN models on the Phy dataset. We also achieve comparable ERR values with the SoTA REDRESS model for the Phy dataset. Moreover, our proposed GFairHint also behaves better than the REDRESS model in most scenarios of the citation network dataset

**Oracle Similarity Matrix based on External Annotation** For the Crime dataset, we construct a discrete (0-1) fairness graph from collected human expert judgements. We show the results for GCN backbone model in Table 2 and the full results with all backbone models in Table 6 (Appendix). We find that for all three backbone GNN models, GFairHint and GFairHint + REDRESS are the best two methods in the fairness (Consistency) evaluation. This is as expected since Vanilla and REDRESS models do not have access to fairness information in this setting. GFairHint and GFairHint + REDRESS have close performance in consistency, demonstrating the effectiveness of fairness hint

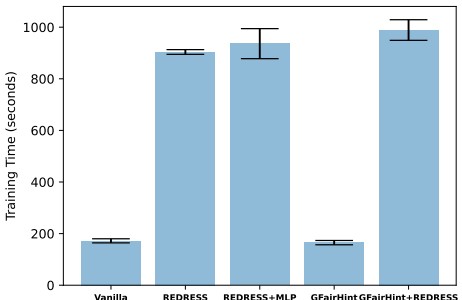

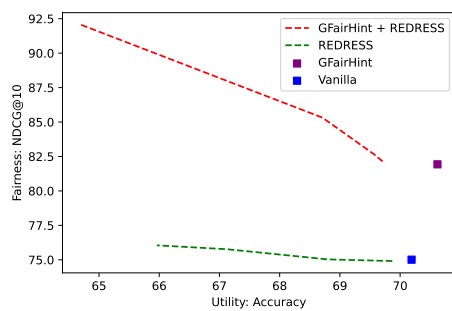

(a) Total time of different models for training 50 epochs.

(b) Utility-fairness tradeoffs for various models with backbone GCN models on Arxiv dataset.

Figure 2: Computation efficiency analysis and Utility-fairness tradeoffs for proposed framework

even when it is used alone. Although GFairHint + REDRESS has slightly better results, it has much higher computational cost because of the ranking-based loss. Detailed discussion on computation efficiency is in Appendix E.2. For GCN and GAT backbone models, our proposed methods achieve the best two results in utility (accuracy) evaluation.

**Summary** We systematically evaluate the utility performance and fairness performance in $5 \times 3 = 15$ combinations of dataset and backbone model, which results in 15 utility comparisons and 27 fairness comparisons.[3]. Our proposed GFairHint + REDRESS method achieved best fairness performance in almost all comparisons (24/27), while GFairHint performed second best in 16/27 of the comparisons when applied alone. These two methods also have comparable utility performance with the Vanilla model, as they ranked top two in 12/15 utility comparisons. Although GFairHint + REDRESS achieved better fairness performance than GFairHint in general, the gaps are small. GFairHint even ranked higher in 6/15 utility performance, especially for large dataset (3/3). These observations empirically show that GFairHint achieves a good balance between utility and fairness, and can perform better when integrated with other complimentary individual fairness promotion methods.

**Efficiency and Sensitivity Analysis** In addition to the main results, we also perform an efficiency and sensitivity analysis of all models. The results are shown in Tables 2a and 2b. For Figure 2a, we observe that the computation cost of GFairHint is comparable with the vanilla model, much less than REDRESS. For Figure 2b, when changing the hyperparameter $\gamma$ to adjust the weight of fairness and utility, our GFairHint + REDRESS model performs consistently better than the original REDRESS. Details of the analysis are shown in the Appendix E.

## 6   Conclusions

In this work, we propose GFairHint, a plug-and-play framework for promoting individual fairness in GNNs via fairness hint. Our method learns fairness hint through an auxiliary link prediction task on a constructed fairness graph. The fairness graph can be derived from both continuous and discrete oracle similarity matrix, corresponding to two ways of obtaining similarity for individual fairness respectively, i.e., from input feature space and from external human annotations. We conduct extensive empirical evaluations on node classification tasks to show the effectiveness of our proposed method in achieving good balance in utility and fairness, with much less computational cost.

One crucial question for individual fairness is the source of similarity measure. External annotation is often impractical, subjective, and potentially biased, and the input feature can be an incomplete and imperfect source. It requires comprehensive domain knowledge to develop the fairness similarity measure for a specific real-world application. We note that our framework is compatible to different types of similarity measure as long as the construction of a fairness graph is viable. Moreover, our

---

[3]For the four academic networks with oracle similarity matrix based on input feature, we evaluated two fairness metrics, which leads to $(4 \times 2 + 1) \times 3 = 27$ comparisons for fairness

framework can be adapted to other tasks in graphs such as link prediction, and other data formats such as text and image. We show the effectiveness of fairness hint with a simple concatenation strategy, and it is possible to develop more complex and better integration methods to utilize fairness hint, especially in compliance with the tasks and data formats at hand. We leave these directions for future work.

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

## A Implementation Details

All the backbone GNN models and our auxiliary link prediction models are implemented in the Pytorch framework, especially the package PyTorch Geometric [30, 11]. For each of our five datasets, we experiment with two backbone GNN settings, small and large model size. For the small model size setting, the number of layers and the dimension of the embeddings in the hidden layers are set to 2 and 16. For the big model size setting, we set these two numbers to 10 and 128 respectively.

For each dataset, we choose model hyperparameter settings with better average utility between two large and small model size settings. Specifically, for citation networks, the results are from 10-layer models with hidden layer dimension 128. For co-authorship netowrks and Crime dataset, the results are from 2-layer models with hidden layer dimension 16. Specifically, for the ArXiv dataset, since they have about 90,000 nodes as the training dataset, using 10-layer GAT on them will cause a memory issue, so we only experiment on the Arxiv data with the 3-layer and 128-dimensional hidden layer GAT model.

For all experiments, we fix the values of the hyperparameters $\gamma$ and $k$ at 1 and 10 as suggested in the previous work [7], where $\gamma$ is the weighting factor when integrating with the ranking-based

loss (Equation 2) and $k$ is the number of top entries used to calculate the ranking loss and fairness evaluation metrics NDCG@K and ERR@K.

Our learned fairness hint for each node contains individual fairness information, and we expect that the fairness hint helps promote fairness in various GNN models. We choose three popular GNN models: GCN [18], GraphSAGE [12], and Graph Attention Networks (GAT) [37] to demonstrate the compatibility of GFairHint with various GNN model designs. Note that we do not need to relearn the fairness hint for the same dataset even if the backbone models have changed.

When training the GCN model for the auxiliary link prediction task, we randomly mask the $2.5\%$ and $5\%$ edges of the fairness graph as the positive edges sampled in the validation set and the test set. We also generate the same number of negative edges. For optimization, we use Adam optimizer with learning rate 0.001 and full batch training [17].

As for the training epochs, when training without the ranking-based loss (Vanilla and FairGraph Embedding models), the numbers of training epochs of ArXiv, ACM, Phy, CS and Crime datasets are 300, 150, 300, 300, 500 respectively. When training with ranking-based loss, we first train the models with only utility loss for tens of epochs to "warm up" and then the models will be trained with ranking-based loss and utility loss together. The numbers of "warm-up" epochs and training epochs with ranking-based loss are 150 and 300, 250 and 300, 50 and 150, 50 and 500, 50 and 600 for ArXiv, ACM, Phy, CS and Crime datasets, respectively. This warm-up operation also follows the procedure in the REDRESS paper [7].

We note that for the Crime dataset, since the entries of oracle similarity matrix $\mathcal{S}_F$ are discrete (0-1), we cannot calculate the ranking-based loss of the constructed fairness graph $\mathcal{G}_F$. Therefore, we adapt the REDRESS-related models to calculate the ranking-based loss based on input feature similarity . As a result, for the Crime dataset, REDRESS and REDRESS + MLP do not have any access to the fairness information (i.e., fairness graph $\mathcal{G}_F$), while GFairHint + REDRESS get fairness information only through the fairness hint but not the ranking-based loss.

## B  Dataset Details

For ACM, CS, and Phy datasets, we follow the preprocessing procedure in REDRESS and use the bag-of-word model to transfer the title and abstract of the paper as node features. We use the pre-split training, validation, and test datasets from the REDRESS paper [4], which samples 5% nodes as the training set, 10% nodes as the validation set, and the rest of the nodes as the test dataset. Regarding the ArXiv dataset, we directly use the processed 128-dimensional feature vectors from a pre-trained skip-gram model [28]. We then follow the train/validation/test splits from the official release of Open Graph Benchmark[5]. We repeat the experiments for each model setting twice, since the split of the dataset is fixed by the previous work.

For the Crime dataset, we have a binary outcome variable for whether the neighborhood is violent and consider other data records as input features. As there is no predefined train/validation/test split, we randomly split the dataset and repeat five times for each model setting. We show basic statistics for the five datasets in Table 3.

| Dataset | # Training Nodes | # Features | # Classes |
|---------|------------------|------------|-----------|
| CS      | 916              | 6,805      | 15        |
| Phy     | 1,724            | 8,415      | 5         |
| ACM     | 824              | 8,337      | 9         |
| ArXiv   | 90,941           | 128        | 40        |
| Crime*  | 1994             | 122        | 2         |

Table 3: Statistics of the datasets used for node classification experiments. * indicates the oracle similarity matrix is discrete and provided by human experts, while the oracle similarity matrix for other datasets are continuous and derived from input feature space.

---

[4]`https://github.com/yushundong/REDRESS/tree/main/node%20classification/data`
[5]`https://ogb.stanford.edu/docs/nodeprop/`

## C Evaluation Metirc Details

ERR@K and NDCG@K measure the similarity between the ranking lists obtained from the oracle similarity matrix $\mathcal{S}_F$ and the outcome similarity matrix $\mathcal{S}_{\hat{Y}}$.

Consistency measures the consistency of outcomes between individuals who are similar to each other. The formal definition regarding a fairness similarity matrix $S_F$ is

$$Consistency = 1 - \frac{\Sigma_i \Sigma_j |y_i - \hat{y_j}| \cdot S_{ij}^F}{\Sigma_i \Sigma_j S_{ij}^F} \qquad \forall i \neq j$$

## D Baseline Details

**Vanilla**: Vanilla denotes the vanilla GNN models without any individual fairness promotion method.

**REDRESS**: REDRESS is the previous SOTA framework for individual fairness promotion in GNN models [7]. They formulate the conventional individual fairness promotion into a ranking-based optimization problem. By optimizing the ranking-based loss $\mathcal{L}_{fairness}$ and the utility loss $\mathcal{L}_{utility}$, REDRESS can achieve the goal of maximization of utility and promotion of individual fairness simultaneously. For the implementation of its framework and ranking-based loss, we adapt the codebase released by the authors[6].

**REDRESS + MLP** As mentioned in Section 3.3, after concatenating the utility node embeddings and fairness hint, our proposed framework GFairHint uses additional MLP layers to process the concatenated embeddings, which increases the model complexity. This variant of REDRESS adds the MLP layers with the same size after the GNN models along with the original REDRESS loss. We use the output of MLP layers from this variation model to calculate the loss and optimize the parameters in the GNN and MLP layers. REDRESS + MLP model can show the effectiveness of GFairHint without interference of the model complexity confounder.

**Our methods:** We study the performance of GFairHint and examine its effectiveness with the combination of REDRESS loss:

**GFairHint**: We combine the fairness hint with the utility node embedding and only use the $\mathcal{L}_{utility}$ loss to update the model parameters.

**GFairHint + REDRESS**: As described in Section 3.4, we combine the ranking-based loss $\mathcal{L}_{fairness}$ in REDRESS with the utility loss $\mathcal{L}_{utility}$ to further encourage the models to learn individual fairness. The only difference between this model and REDRESS + MLP is that GFairHint + REDRESS incorporates the fairness hint.

## E Efficiency and Sensitivity Analysis

### E.1 Trade-off between Fairness and Utility

GFairHint + REDRESS achieves the best fairness performance, where we integrate fairness hint with ranking-based loss. The value of the hyperparameter $\gamma$ in Equation 2 controls the strength of the fairness constraint. There is a trade-off between utility and fairness when adjusting the $\gamma$ value [7]. To demonstrate the effectiveness of GFairHint, we perform experiments with multiple values of $\gamma$ for the REDRESS and GFairHint + REDRESS models on the Arxiv dataset with GCN as the backbone GNN model. Figure 2b shows the trade-off between accuracy and fairness (NDCG@10) with varying values of $\gamma$ for the REDRESS and GFairHint + REDRESS methods. The curves in the figure for REDRESS and GFairHint + REDRESS are generated by changing a range of fairness coefficients $\gamma$ from 0.01 to 100 and computing the Pareto frontiers. We also further visualize the accuracy and NDCG@10 values for the vanilla and GFairHint models as two data points for reference.

With the value of $\gamma$ being small (e.g., 0.001), REDRESS and GFairHint + REDRESS models behave similarly to the vanilla and GFairHint models respectively as expected. When increasing the value of $\gamma$, we can observe fairness improvements for both REDRESS and the GFairHint + REDRESS models. This fairness improvement is more significant for the GFairHint + REDRESS model. We conjecture

---

[6]https://github.com/yushundong/REDRESS

that little improvement for the REDRESS model is due to the vanishing gradient problem of deep GNN models [5], which may reduce the impact of fairness loss. GFairHint + REDRESS model is less impacted because it also directly learns from the fairness hint that is incorporated into the final GNN layers. We observe that with the same accuracy level, our proposed GFairHint + REDRESS model achieves a higher NDCG@10 value than the REDRESS model, demonstrating the effectiveness of the fairness hint.

We expect that the adjustment of the trade-off between fairness and utility can provide more flexibility in practical applications. For example, some tasks may pay more attention to fairness rather than utility.

## E.2 Efficiency Evaluation

In addition to fairness and utility results, we compare the efficiency of GFairHint models with other baseline models in terms of time complexity and empirical training time. We do not count the training and inference time of fairness representation learning model, since the generated fairness hint can be re-used by different backbone GNN models for multiple times. During the training phase, the ranking-based loss in REDRESS requires to find a list containing top-k similar nodes for each node and rank the list. For the time complexity of the rank-based loss, we need to find the top-k similar nodes for each node, so the time complexity is at least $\mathcal{O}(n \cdot \log(n) \cdot k)$. For our GFairHint model, the additional computation focuses on two MLP layers and the time complexity is $\mathcal{O}(n)$. Therefore, the training time of REDRESS is much longer than that of the vanilla GNN models. We claim that the GFairHint framework has a much lower computation cost than REDRESS does. To show the gap more clearly, we perform experiments on the largest dataset, ArXiv, which has 90k training nodes. We choose GCN as the backbone model. The experiments were conducted in a controlled computation environment with single GPU (RTX2080ti) and fixed GPU memory (32GB). For each method, we train the models for 300 epochs and visualize the average training time of 50 epochs in Figure 2a. The training time of GFairHint is significantly shorter than the time of other fairness methods, and the additional cost is ignored compared to the Vanilla model. Therefore, the proposed GFairHint model is more scalable in pratice when applied to large graph datasets.

| Dataset | BB | Model | ACC | ERR@10 | NDCG@10 |
|---------|----|----|-----|--------|---------|
| **ArXiv** | GCN | Vanilla | 70.19 ± 0.02 | 91.45 ± 0.01 | 75.01 ± 0.19 |
| | | REDRESS | 68.65 ± 1.13 | 91.57 ± 0.10 | 75.25 ± 0.71 |
| | | REDRESS + MLP | 69.85 ± 0.27 | 91.47 ± 0.01 | 75.01 ± 0.26 |
| | | GFairHint | **70.62 ± 0.91** | 94.28 ± 0.09 | 81.93 ± 0.27 |
| | | GFairHint + REDRESS | 69.80 ± 0.47 | **95.22 ± 0.80** | **85.48 ± 3.47** |
| | SAGE | Vanilla | **70.44 ± 0.69** | 91.71 ± 0.13 | 75.47 ± 0.37 |
| | | REDRESS | 69.34 ± 0.55 | 91.58 ± 0.16 | 75.51 ± 0.73 |
| | | REDRESS + MLP | 69.75 ± 0.18 | 91.40 ± 0.09 | 74.45 ± 0.53 |
| | | GFairHint | 70.40 ± 0.34 | 94.34 ± 0.04 | 81.92 ± 0.17 |
| | | GFairHint + REDRESS | 68.98 ± 0.25 | **95.22 ± 0.79** | **85.32 ± 3.45** |
| | GAT | Vanilla | 70.86 ± 0.64 | 92.04 ± 0.09 | 76.64 ± 0.21 |
| | | REDRESS | 69.74 ± 0.19 | 92.18 ± 0.02 | 77.46 ± 0.09 |
| | | REDRESS + MLP | 70.45 ± 0.30 | 91.86 ± 0.25 | 76.23 ± 0.98 |
| | | GFairHint | **71.06 ± 0.45** | 94.20 ± 0.04 | 81.80 ± 0.14 |
| | | GFairHint + REDRESS | 69.89 ± 0.11 | **95.20 ± 1.19** | **85.49 ± 4.73** |
| **ACM** | GCN | Vanilla | **70.78 ± 0.18** | 76.99 ± 0.08 | 33.90 ± 0.73 |
| | | REDRESS | 70.15 ± 1.77 | 76.98 ± 0.13 | 34.82 ± 0.80 |
| | | REDRESS + MLP | 70.64 ± 1.89 | 76.66 ± 0.19 | 30.93 ± 0.46 |
| | | GFairHint | 69.70 ± 0.77 | 76.39 ± 0.52 | 35.12 ± 0.34 |
| | | GFairHint + REDRESS | 69.77 ± 0.95 | **77.00 ± 0.16** | **38.58 ± 2.85** |
| | SAGE | Vanilla | 69.26 ± 0.60 | 76.63 ± 0.18 | 30.55 ± 1.86 |
| | | REDRESS | 68.23 ± 0.97 | 76.68 ± 0.04 | 31.58 ± 1.06 |
| | | REDRESS + MLP | **69.32 ± 0.44** | 76.29 ± 0.78 | 28.73 ± 0.12 |
| | | GFairHint | 69.24 ± 0.11 | 76.39 ± 0.25 | 36.12 ± 0.72 |
| | | GFairHint + REDRESS | 67.52 ± 0.16 | **77.37 ± 0.55** | 37.83 ± 3.78 |
| | GAT | Vanilla | **71.14 ± 1.14** | 77.00 ± 0.20 | 34.62 ± 0.28 |
| | | REDRESS | 70.49 ± 0.87 | 77.40 ± 0.28 | 34.83 ± 0.45 |
| | | REDRESS + MLP | 69.87 ± 0.70 | 76.22 ± 0.09 | 32.82 ± 1.36 |
| | | GFairHint | 71.04 ± 0.74 | 76.79 ± 0.27 | 37.52 ± 0.54 |
| | | GFairHint + REDRESS | 69.65 ± 0.88 | **77.50 ± 0.36** | **43.01 ± 2.02** |

Table 4: Node classification results for citation datasets: ArXiv and ACM. BB represents the backbone GNN models. The number of layers and the hidden layer dimension of the backbone GNN models are 10 and 128 respectively. All values are reported in percentage. The Best results are in **bold**, the second best results are underlined.

| Dataset | BB | Model | ACC | ERR@10 | NDCG@10 |
|---|---|---|---|---|---|
| | | Vanilla | 80.16 ± 9.32 | 78.93 ± 0.07 | 44.00 ± 1.14 |
| | | REDRESS | 79.88 ± 2.68 | 81.25 ± 0.55 | 49.24 ± 2.36 |
| | GCN | REDRESS + MLP | 77.35 ± 2.10 | 78.76 ± 0.27 | 40.54 ± 2.45 |
| | | GFairHint | 87.08 ± 2.05 | 79.56 ± 0.36 | 51.31 ± 1.17 |
| | | **GFairHint + REDRESS** | **91.17 ± 0.54** | **83.48 ± 0.20** | **64.60 ± 0.58** |
| | | Vanilla | 85.49 ± 6.58 | 78.33 ± 0.02 | 46.34 ± 0.98 |
| | | REDRESS | 88.26 ± 3.30 | 81.09 ± 0.59 | 54.71 ± 1.91 |
| CS | SAGE | REDRESS + MLP | 83.29 ± 1.66 | 77.80 ± 0.44 | 42.83 ± 1.40 |
| | | GFairHint | 86.67 ± 3.07 | 79.32 ± 0.30 | 51.00 ± 0.73 |
| | | **GFairHint + REDRESS** | **91.06 ± 0.02** | **83.21 ± 0.31** | **64.49 ± 0.45** |
| | | Vanilla | 80.73 ± 7.52 | 79.44 ± 0.29 | 46.99 ± 0.98 |
| | | REDRESS | 79.53 ± 2.75 | 80.62 ± 0.00 | 51.14 ± 0.25 |
| | GAT | REDRESS + MLP | 82.42 ± 1.87 | 79.13 ± 0.48 | 44.39 ± 0.73 |
| | | GFairHint | 86.11 ± 0.94 | 80.91 ± 0.38 | 53.80 ± 0.99 |
| | | **GFairHint + REDRESS** | **90.54 ± 0.57** | **83.46 ± 0.33** | **63.67 ± 0.09** |
| | | Vanilla | 88.33 ± 5.11 | 73.30 ± 0.10 | 30.46 ± 1.05 |
| | | REDRESS | 84.28 ± 2.12 | **74.69 ± 0.06** | 35.76 ± 1.72 |
| | GCN | REDRESS + MLP | 93.40 ± 0.38 | 74.61 ± 0.13 | 36.06 ± 0.88 |
| | | GFairHint | 87.35 ± 0.03 | 71.60 ± 0.32 | 33.26 ± 0.21 |
| | | **GFairHint + REDRESS** | **94.15 ± 0.15** | 73.87 ± 1.46 | **41.53 ± 4.63** |
| | | Vanilla | **95.65 ± 0.51** | 72.38 ± 0.33 | 31.73 ± 0.04 |
| | | REDRESS | 89.72 ± 0.33 | **74.89 ± 0.64** | 41.01 ± 2.25 |
| Phy | SAGE | REDRESS + MLP | 93.08 ± 0.04 | 74.22 ± 0.04 | 35.65 ± 0.90 |
| | | GFairHint | 90.00 ± 3.15 | 71.75 ± 0.28 | 29.95 ± 0.98 |
| | | GFairHint + REDRESS | 93.24 ± 0.69 | 74.06 ± 0.03 | **41.66 ± 0.23** |
| | | Vanilla | 90.33 ± 5.19 | 73.78 ± 0.57 | 33.27 ± 1.32 |
| | | REDRESS | 84.74 ± 5.39 | 74.64 ± 0.34 | 36.24 ± 0.06 |
| | GAT | REDRESS + MLP | 92.54 ± 0.15 | **74.70 ± 0.97** | 36.43 ± 2.78 |
| | | GFairHint | 89.79 ± 1.98 | 71.99 ± 0.00 | 29.13 ± 0.25 |
| | | **GFairHint + REDRESS** | **93.67 ± 0.30** | 74.62 ± 1.10 | **44.56 ± 2.62** |

Table 5: Node classification results on co-authorship datasets: coauthor-phy and coauthor-cs. BB represents the backbone GNN models. The number of layers and the hidden layer dimension of backbone GNN models are 2 and 16 respectively. All values are reported in percentage. The best results are in **bold**, the second best results are underlined.

| BB | Model | ACC | Consistency |
|---|---|---|---|
| | Vanilla | 73.83 ± 0.34 | 54.80 ± 0.23 |
| | REDRESS | 73.98 ± 0.70 | 54.07 ± 0.96 |
| GCN | REDRESS + MLP | 73.58 ± 1.80 | 53.06 ± 1.04 |
| | GFairHint | 75.44 ± 0.71 | 62.76 ± 2.74 |
| | **GFairHint + REDRESS** | **75.54 ± 0.90** | **63.61 ± 4.44** |
| | Vanilla | **82.16 ± 0.33** | 62.09 ± 0.50 |
| | REDRESS | 82.11 ± 0.52 | 61.46 ± 1.91 |
| SAGE | REDRESS + MLP | 81.35 ± 0.34 | 61.46 ± 1.36 |
| | GFairHint | 80.60 ± 0.98 | 62.26 ± 0.98 |
| | GFairHint + REDRESS | 80.85 ± 1.21 | **62.49 ± 4.86** |
| | Vanilla | 73.68 ± 0.79 | 55.17 ± 0.81 |
| | REDRESS | 72.88 ± 0.74 | 53.55 ± 1.15 |
| GAT | REDRESS + MLP | 72.08 ± 1.24 | 51.84 ± 0.42 |
| | **GFairHint** | **75.34 ± 0.74** | 64.04 ± 2.74 |
| | GFairHint + REDRESS | 74.94 ± 1.05 | **65.30 ± 3.60** |

Table 6: Node classification results on the Crime datasets. BB represents the backbone GNN models. The number of layers and the hidden layer dimension of backbone GNN models are 2 and 16 respectively. All values are reported in percentage. The best results are in **bold**, the second best results are underlined.

