# OpenReview forum: "GFairHint: Improving Individual Fairness for Graph Neural Networks via Fairness Hint"
_NeurIPS.cc/2022/Workshop/TSRML — TSRML2022_

### Official Review · Reviewer_umHR · 2022-10-20
**A trivial paper, method is trivial and the results are not impressive**

**Overall Rating:** 4

**Summary:**

In this paper, the authors proposed a method to improve individual fairness in GNNs. Besides using the original graph to get node embeddings. The authors also generated edges by similarity matrix S_F, and get node embeddings from the new graph G_F. Then concatenate the 2 embeddings together followed by an MLP.
The method is lack innovation, the idea of the Oracle similarity matrix is borrowed from other papers, not their own idea.


**Strengths:**

The motivation of this paper is valuable. And the authors reported different GNN structures + their fairness methods.

**Weaknesses:**

1 the authors' method does not perform constantly across datasets and GNN models. Sometimes it is better than the base. Sometimes it is worse. We did not see a significant performance gain in the metrics.

2 Their GFairHint method is mainly putting 2 embeddings generated from different graphs together, and the way to generate the links is not their own contributions.


**Overall Recommendation:**

Although the motivation for this paper is good, it lacks innovations and the result did not show significant benefit with their GFairHint method.

**Review Confidence:**

3: The reviewer is fairly confident that the evaluation is correct

---

### Official Review · Reviewer_o9kf · 2022-10-21

**Overall Rating:** 7

**Summary:**

The authors study the important problem of Fairness in GNN. Standard GNN methods ignore notions of fairness like the notions of equality such as that similar nodes should receive similar treatment. The paper address this problem with an heuristic method that consists in providing a graph of hints for fairness. The hint graph is a set of edges of like-wise nodes that should be treated similarly. By providing the embeddings of likewise nodes v(u) in training for a node u, the algorithm can make fairer decisions. The authors verify this with an empirical evaluation.

**Strengths:**

Good empirical evidence of improvements of fairness.

**Weaknesses:**

Lack of theoretical validation for the model.

**Overall Recommendation:**

The papers appears to be sufficiently interesting to be present at the workshop.

**Review Confidence:**

2: The reviewer is willing to defend the evaluation, but it is quite likely that the reviewer did not understand central parts of the paper

---

### Decision · Program_Chairs · 2022-10-23

**Decision:**

Accept

**Comment:**

Following the recommendation from the review, the submission is accepted. We encourage the authors to improve the manuscript, especially by clarifying the metrics that correspond to fairness in Tables 1 and 2 to avoid confusion, in the camera-ready version.